cognition/health and disease and epidemiology/behaviour

coronavirus, COVID-19, risk perception, protective behaviour, pandemic

**Authors for correspondence:**
Toby Wise
e-mail: tobywise@caltech.edu
Dean Mobbs
e-mail: dmobbs@caltech.edu

# Changes in risk perception and self-reported protective behaviour during the first week of the COVID-19 pandemic in the United States

Toby Wise[1,2,3], Tomislav D. Zbozinek[1], Giorgia Michelini[4], Cindy C. Hagan[1] and Dean Mobbs[1,5]

[1]Division of Humanities and Social Sciences, California Institute of Technology, Pasadena, CA, USA
[2]Max Planck UCL Centre for Computational Psychiatry and Ageing Research, and [3]Wellcome Centre for Human Neuroimaging, University College London, London, UK
[4]Semel Institute for Neuroscience and Human Behavior, University of California Los Angeles, Los Angeles, CA, USA
[5]Computational Neural Systems Program, California Institute of Technology, Pasadena, CA, USA

TW, 0000-0002-9021-3282

Efforts to change behaviour are critical in minimizing the spread of highly transmissible pandemics such as COVID-19. However, it is unclear whether individuals are aware of disease risk and alter their behaviour early in the pandemic. We investigated risk perception and self-reported engagement in protective behaviours in 1591 United States-based individuals cross-sectionally and longitudinally over the first week of the pandemic. Subjects demonstrated growing awareness of risk and reported engaging in protective behaviours with increasing frequency but underestimated their risk of infection relative to the average person in the country. Social distancing and hand washing were most strongly predicted by the perceived probability of personally being infected. However, a subgroup of individuals perceived low risk and did not engage in these behaviours. Our results highlight the importance of risk perception in early interventions during large-scale pandemics.

## 1. Introduction

The genesis of COVID-19 has been tied to the Hubei province of China, and it rapidly progressed from local epidemic to the

level of a global pandemic, with countries across the globe reporting exponentially increasing numbers of infections and deaths [1]. The first US case was reported on 14 January 2020 [2], followed by government interventions in travel restrictions. On 11 March, COVID-19 officially become an global pandemic [3], and the introduction of governmental directions to restrict social and economic behaviour began in the United States. By 17 March 2020, all 50 US states reported at least one person with the virus [2]. A major focus of authorities in the United States, and in other countries, has been to minimize transmission of the virus in order to flatten the epidemic peak and lessen the impact on healthcare services [4,5]. This is critical in the case of COVID-19 due to its high transmissibility—even in the absence of symptoms [6,7]—combined with its severity [4] and mortality rate, particularly among older individuals [5]. However, these measures rely on rapid changes in population behaviour, which are dependent on individuals' ability to perceive risks associated with the virus and adapt their behaviour accordingly [8].

It is, therefore, crucial to assess psychological and behavioural responses to the pandemic and determine how perceived risk is linked to engagement in protective behaviours [9]. There is limited evidence on psychological and behavioural responses in the early stages of prior pandemics when preventative measures are most necessary [9]. While some studies have emphasized the role of perceived risk to the self [9], these have rarely been conducted during the emergence of an outbreak. The few studies that have surveyed individuals during the early stages of a pandemic have also suggested that perceived personal risk of infection and health effects are linked to engagement in protective behaviours [10]. By contrast, experimental psychology research has suggested that increased perceived effects of disease spread on others may increase engagement in social distancing [11]. However, it is also well established that individuals underestimate their probability of experiencing adverse life events (such as cancer) relative to the average person—an effect known as optimism bias [12]. Together, these lines of research suggest that perceived risk is likely to affect individuals' behaviour during a pandemic but that individuals are often poor at perceiving risk. However, it is unknown how perceived risk relates to protective behaviours in the early stages of a pandemic on the scale of COVID-19. More importantly, we are unaware of any data of this kind for the current COVID-19 pandemic. Given COVID-19's ongoing rampant nature and the need to enact protective behaviours at a population level [4], such data may have global value for countries where the virus has not yet spread and for future pandemics.

The current study, therefore, sought to investigate perceptions of risk and behavioural responses to the pandemic in individuals from the United States during the early stages of the outbreak in March 2020.

# 2. Material and methods

## 2.1. Study design

The study was a combined cross-sectional and longitudinal survey. The cross-sectional component consisted of surveys completed by different subjects on 11, 12, 13, 14 and 16 March 2020 to allow between-group comparison of results over time. The longitudinal component involved following up the first cohort of subjects (responding on 11 March 2020) 5 days later (15 March 2020), although some subjects completed the survey the following day (16 March 2020). This enabled us to confirm that any differences across the groups recruited on different days were truly longitudinal changes, rather than being a result of differences between subjects tested on different days.

Although the 5-day follow-up period is short, many global events occurred during this time (electronic supplementary material, figure S1), including travel bans and restrictions on public gatherings. Notably, testing occurred before and after the recommendation by the CDC (Centers for Disease Control and Prevention, the United States public health institute) of avoiding gatherings of 50+ people (15 March 2020) [13] and on the same day of President Trump's recommendation to avoid gatherings of 10+ people (16 March 2020) [14].

## 2.2. Subjects

We recruited 1591 subjects through Prolific [15], in groups of 502, 273, 253, 259 and 299 on each of the days given above. Another five participants participated but were not included in any analyses involving time due to a technical error in recording date information. Of the 502 subjects recruited on

the first day, 495 who provided complete data were invited to recomplete the survey 5 days later. Of these, 375 completed the survey, representing a 76% retention rate. For longitudinal analyses, we compared data from this follow-up time point (referred to as T2) with data from these same subjects collected on 11 March 2020 (referred to as T1).

Subjects were young on average, with a median age of 30. Fifty-five per cent of subjects were female, 40% were male and 5% did not report sex. In terms of employment, 13.1% reported being unemployed and job seeking, 22.16% were in part-time employment, 35.78% were in full-time employment, 10.67% reported not being in paid work (being a homemaker, retired or disabled), 6.59% indicated that they did not fall into these categories and 10.29% provided no data. A total of 64.09% of the sample indicated that they were currently a student, 30.00% indicated they were not and 5.9% provided no answer to this question. When asked for the highest level of education they had achieved, 1.39% reported completing some high school, 40.15% had completed high school, 39.5% had a bachelor's degree, 13.69% had a master's degree, 3.7% held a PhD or higher and 1.57% did not respond. Demographic information is shown in electronic supplementary material, figure S2.

Data on location at the state level were recorded for 948 subjects (these data were not recorded for the remaining subjects due to a technical error). This revealed that subjects were distributed across the country as would be expected based on state population, as demonstrated by a strong correlation between the number of responses from each state and the estimated 2019 population [16] of the state ($r = 0.98$, $p < 0.001$, electronic supplementary material, figure S2).

## 2.3. Questionnaires

Questions were designed to ask subjects primarily about their perception of risk from COVID-19 and their behaviour in response to it (further details for each question are available in electronic supplementary material, table S1). Risk perception questions asked about the perception of risk in terms of infection likelihood and severity and asked about multiple hypothetical 'average' people in addition to the subject themselves (for example the average person in the neighbourhood, state and country). We used the term 'likelihood' in the items as it is colloquially interpreted as representing probability; however, we assume that responses refer to the probability of the event occurring in the formal statistical sense. We also asked about the perceived risk of transmitting the infection to another person, and how badly this person would be affected, based on research suggesting that these factors influence the adoption of social distancing behaviour [17]. Protective behaviour questions focused on behaviours known to slow the spread of disease, primarily hand washing and social distancing (reflected in a tendency to see fewer people, stay at home more and travel less). We also included questions about purchasing extra household supplies and food given the increase in 'panic' buying reported by news outlets at the time.

We also questioned people about how badly they felt they had been personally affected by the pandemic and asked about the extent to which they were seeking information online. We also asked whether they had been avoiding news due to anxiety.

All items were rated using a visual analogue scale ranging from 0 to 100, with anchors at either end depending on the question asked. Subjects positioned a movable slider to indicate their response, and the slider only became visible after clicking on the scale to prevent subjects from being biased in their response by its original position.

## 2.4. Statistical analysis

### 2.4.1. Descriptive statistics

Descriptive statistics were calculated using means and standard deviations, except for behavioural items, which exhibited non-normal distributions, and age, which was positively skewed. In this case, we report medians and interquartile ranges.

### 2.4.2. Changes in risk perception and self-reported engagement in protective behaviours over time

We tested for changes over time in two ways: first, we used mixed ANOVAs to test for differences in responses across samples collected on different dates. For analyses focusing on risk perception, the dependent variable was the perceived probability of infection. The within-subject factor was the subject of the rating (self, average person in the neighbourhood, average person in the state, average

person in the country), and the between-subject factor was the date of testing. For analyses focusing on behaviour, our dependent variable was the likelihood of attending an event with a specific number of other people (10, 50, 100, 500 or 1000), with the within-subject factor being the number of people and the between-subject factor being the date of testing.

Second, we used repeated ANOVAs in our longitudinal dataset to test for within-subject changes over time. Here, the factors were the same as above; however, date of testing was a within-subject factor in this case. This allowed us to confirm whether differences in samples tested across different days did indeed represent a true change over time.

All of the analyses related to changes over time were determined *a priori*, and we hypothesized that perceived risk would increase over time, while willingness to attend events with other people would decrease over time. We also expected subjects to rate infection probability as being higher for others than for themselves and expected to see lower willingness to attend events with higher numbers of people.

### 2.4.3. Predictors of self-reported engagement in protective behaviours

For our analyses of predictors of engagement in protective behaviours, we used a multiple linear regression model, entering all measures of risk perception as predictors, in addition to age as a covariate. For outcome variables, we used items measuring the extent to which subjects were washing their hands and staying home, as indicators of sanitization and social distancing strategies, respectively. We also performed additional analyses using optimism bias, calculated as the difference between perceived infection probability for the self versus that for the average person in the state, as a predictor in addition to age as a covariate. While we did ask questions relating to a wider range of protective behaviours, we elected to limit our analyses to these two measures to reduce the complexity of our analyses and avoid testing large numbers of variables for potential associations. These two items were chosen as they represented behaviours that were known to help reduce the spread of the virus at the time.

We elected not to include sex as a covariate due to a relatively high number of missing responses to this question; however, the results we report did not change when excluding these subjects and including sex as an additional covariate. All predictors were scaled to zero mean and unit variance prior to analysis to allow comparability of regression coefficients. These analyses were exploratory, and so we first performed this analysis in a discovery dataset consisting of 75% of subjects' data, randomly selected. We then ran the same analysis in the remaining 25% to ensure the replicability of our results.

We checked for multicollinearity between predictor variables using the variance inflation factor. All values were below 5, indicating no problematic multicollinearity between predictors [18].

### 2.4.4. Identification of subgroups

To decompose responses to the four items relating to protective behaviours and identify clusters within them, we used a Bayesian Gaussian mixture modelling approach. This approach seeks to decompose the multivariate distribution (such as that of the responses to the four questions) into a mixture of Gaussian distributions. This analysis was entirely exploratory, we did not have *a priori* hypotheses regarding the subgroups and instead allowed these to be derived from the data. For this data-driven clustering analysis, we used four measures of engagement in protective behaviours (avoiding social interaction, hand washing, staying home and travelling less) as using multiple indicator variables provides more information from which to identify clusters.

An advantage of the Bayesian approach is that it is able to select an appropriate number of components; by estimating the weight of the components, unused components can be set to have negligible weight and hence were excluded from the model. This analysis was conducted using the implementation available in Scikit-Learn [17]. We used a Dirichlet distribution, representing a finite mixture, as the prior on component weights and set the covariance type to full. We set the maximum number of components to 20 and rejected any with a weight below 0.01 as these had a negligible contribution to the model. This left 16 components as the final solution.

To examine how subjects in a group characterized by low reported engagement in protective behaviours differed from the average subject, we first calculated z-scores on a number of variables of interest (related to risk perception and effects of the pandemic). We followed this by testing which of

these variables predicted membership of the low engagement group versus the rest of the sample using logistic regression.

### 2.4.5. Demographic effects on risk perception and self-reported engagement in protective behaviour

We also explored effects of age, sex and location on risk perception and engagement in protective behaviours. To examine the effects of age, we split subjects into three groups based on age (18–26, 26–35, 35–80) using a tertile split. For analyses involving location, we assigned each individual to a census-defined region according to their state (https://www.census.gov/geographies/reference-maps/2010/geo/2010-census-regions-and-divisions-of-the-united-states.html), resulting in each subject being allocated to one of the following regions: South, West, Midwest or North East. Only subjects where location data were available were included in this analysis.

We used ANOVAs to examine the effects of group on responses in the case of age and location, and *t*-tests in the case of sex, and education level, with *p*-values corrected for multiple comparisons using false discovery rate (FDR) correction. Full results for these analyses are provided in electronic supplementary material, tables S1–S8 and figures S3–S10.

# 3. Results

We conducted an online study of subjects recruited through Prolific [15] over 5 days between 11 March 2020, the day when the WHO declared COVID-19 a pandemic, and 16 March 2020 (see electronic supplementary material, figure S2 and Supplementary Results for demographic information). The geographical distribution of subjects was consistent with state populations (figure 1a,b), indicating that the sample was geographically representative.

## 3.1. Perceptions of risk from COVID-19

Although it is difficult to determine exactly how widespread the pandemic will be in the United States, estimates reported in the media when data were collected suggested that up to 80% of the population may contract the disease [4]. We investigated perceptions of infection probability and severity for both the study participants themselves and others on a scale from 0 to 100.

As shown in figure 1c, subjects assessed their risk of being infected as relatively high (mean = 43.06, s.d. = 26.62). Additionally, they reported perceiving the disease as being a threat to their health (mean = 44.70, s.d. = 26.93), indicated that they would be personally affected economically, such as through loss of work (mean = 45.68, s.d. = 34.35), and that they would be affected by the global economic consequences, such as through economic recession and effects on healthcare provision (mean = 64.38, s.d. = 24.02).

Subjects were also aware of the potential for contagion, indicating that if they became infected, they would be likely to pass it to someone else (mean = 66.18, s.d. = 27.39). Subjects believed that if they did infect another person, this person would be more severely affected than themselves in terms of health (mean difference = 14.82, s.d. difference = 26.67). Given that the sample was relatively young (median age = 30 years), it is possible that this is an artefact of the young age of participants combined with the knowledge that the virus affects older individuals more severely. We tested this by predicting the difference in ratings between self and other from age. Linear regression indicated that the difference between perceived effects on another person and reported personal health risk was partially dependent on age ($t_{1550} = -8.33$, $p < 0.001$). However, the intercept in this model remained positive and significant ($\beta = 29.68$, $p < 0.001$), indicating the presence of such a bias even after accounting for age.

The perceived probability of infection differed according to who participants were rating ($F_{3,4737} = 579.00$, $p < 0.001$, $\eta_p^2 = 0.27$), with participants rating the average person in the United States to have the highest risk of infection, but themselves to have the lowest risk. This effect can be interpreted as an optimism bias [12] (figure 2a), whereby subjects perceive themselves at lower risk than the average person. In between-subject analyses, the perceived probability of infection differed across samples tested on different days, demonstrating a higher rate in subjects tested on later dates ($F_{6,1579} = 6.48$, $p < 0.001$, $\eta_p^2 = 0.024$, figure 2a). To confirm this represented a change over time, we examined increases in perceived probability within-subjects in a subsample followed up after 5 days, finding that this did indeed increase over time ($F_{1,374} = 69.19$, $p < 0.001$, $\eta_p^2 = 0.16$, figure 2b). In this within-subjects analysis, there was an interaction between time and the person being rated ($F_{3,1122} = 7.56$, $p < 0.001$, $\eta_p^2 = 0.02$), with the greatest changes over time emerging in risk perception for the self

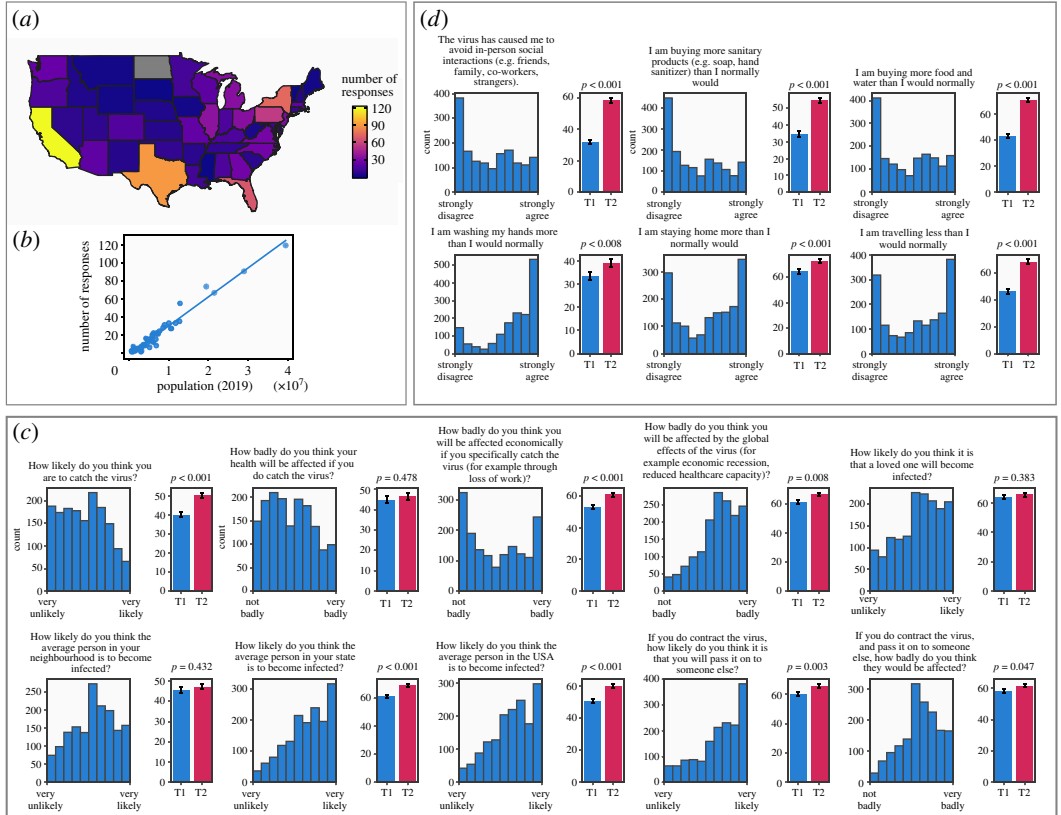

**Figure 1.** Location of subjects and distributions of responses to items regarding risk perception and protective behaviour ($n = 1591$). (*a*) Number of responses in each state in the mainland United States. (*b*) Correlation between number of responses and state population, indicating that the number of responses was in line with expectations based on population. (*c*) Distribution of responses to risk perception items and (*d*) distribution of responses to protective behaviour items. All responses were recorded on a visual analogue scale ranging from 0 to 100. Bar plots indicate mean responses to these items over the two time points where a subgroup of subjects was retested ($n = 375$), and *p*-values represent results from a repeated measures *t*-test between the two time points. Subject-level changes over the two time points are shown in electronic supplementary material, figures S9 and S10.

(as opposed to other people); however, this was weak and potentially influenced by ceiling effects given the high initial ratings for others. Together, these results indicate a clear pattern of optimistic risk perception, with subjects rating themselves as being at lower risk of infection than the average person, which changed rapidly over the period of the study.

It is possible that the observed optimism bias here may reflect the relatively low risk level of the sample, potentially being at genuinely lower risk than the average person in their state or country. To assess the influence of these factors, we used an ANOVA predicting optimism bias (the difference between perceived risk for the self and that for the average person in the state) from age, sex and location. This revealed a significant, albeit weak, negative effect of age ($F_{1,873} = 5.87$, $p = 0.016$). The effects of sex and location were not significant ($ps = 0.77$ and $0.65$, respectively). We followed this by using a linear regression predicting optimism bias scores from age alone to determine the proportion of variance explained by age. This resulted in an $R^2$ of 0.005, indicating that only 0.5% of the variance in optimism bias scores was explained by age.

We observed limited effects of demographic variables on risk perception. Full details of these analyses, including results of all statistical tests, are provided in the electronic supplementary material, but we summarize the main findings here. Younger adults believed they were less likely to become infected and would be less severely ill if they did become infected, while females believed they were more likely to pass on the virus if infected than males. There were no effects of subjects' geographical location of residence. We observed the strongest effects when focusing on education level, while controlling for age. Individuals who had completed college-level education or higher reported their personal probability and severity of infection, along with that of the average person in their neighbourhood, as higher than those who had not completed college-level education.

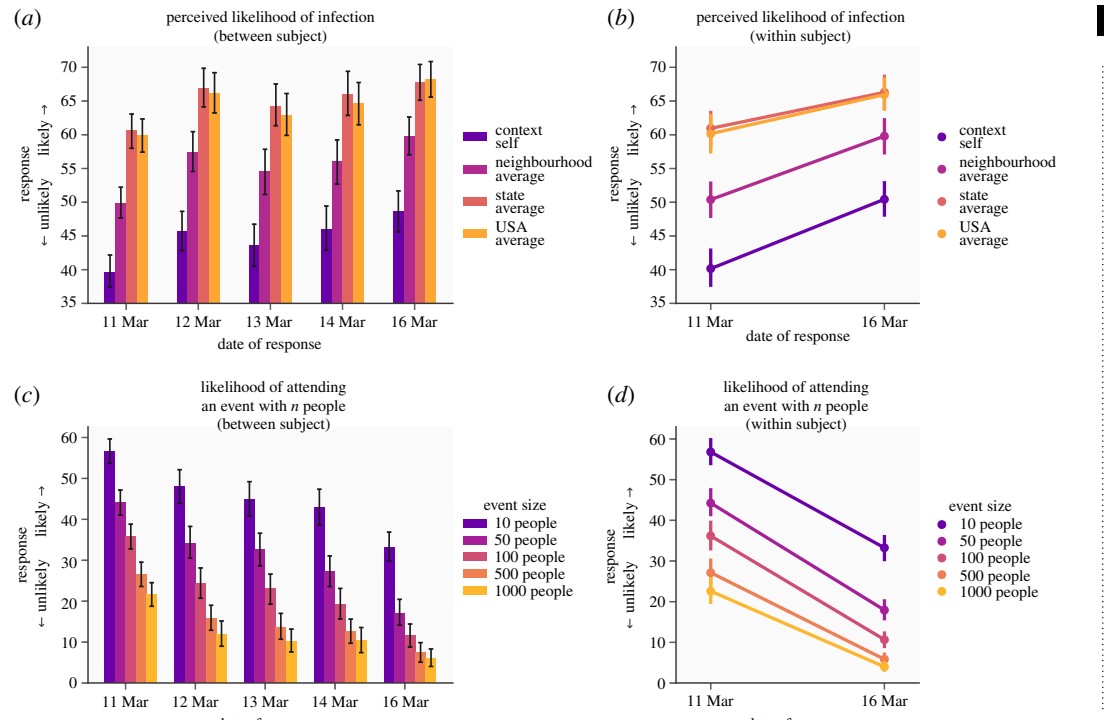

**Figure 2.** Changes in protective behaviours and risk perception over time. (*a*) Perceived probability of becoming infected for participants themselves and average people at different geographical scales in separate samples tested over 5 days. (*b*) Perceived likelihoods of infection in a subset of subjects followed up after 5 days. (*c*) Reported likelihood of attending events with a given number of other people in separate samples tested on 5 days in the early stages of the outbreak in the United States. (*d*) Reported probability of attending events of different sizes in a subset of subjects followed up 5 days after initially completing the survey.

## 3.2. Self-reported engagement in protective behaviours

We next assessed the extent to which subjects reported engaging in protective behaviours, such as social distancing and hand washing, in addition to superficially helpful behaviours such as buying more food and water. On average, subjects indicated that they were engaging in such behaviours more than usual, although response distributions included peaks at the extremes (figure 1*d*). Subjects reported washing their hands more than normal (median = 77, IQR = 38) and staying home more than they would usually (median = 62, IQR = 69), indicating high engagement with sanitization and social distancing measures. In subjects who completed the survey at a second time point 5 days after the first completion (16 March 2020), responses had changed for both hand washing (Wilcoxon $W_{375} = 25027.5$, $p < 0.001$) and social distancing ($W_{375} = 12269$, $p < 0.001$), reflecting increased within-person engagement in these behaviours (figure 1*d*).

We also asked people how likely they would be to attend events with varying numbers of people (10–1000) to assess how they were adapting their behaviour according to transmission risk. As expected, we observed a main effect of group size ($F_{4,6316} = 1311.68$, $p < 0.001$, $\eta_p^2 = 0.45$, figure 2*c*), whereby individuals were less likely to attend an event with more people. In between-subject analyses, we also saw markedly lower probability ratings over time in separate samples collected across multiple days ($F_{6,1579} = 22.84$, $p < 0.001$, $\eta_p^2 = 0.08$, figure 2*c*). Congruently, a decrease over time emerged in our within-subject analysis ($F_{1,374} = 279.02$, $p < 0.001$, $\eta_p^2 = 0.43$, figure 2*d*), providing direct evidence that individuals reported dramatically changing their intended behaviour within the space of only a few days. Together, these results indicate that subjects generally reported engaging in protective behaviours, and that their level of engagement increased dramatically over in the 5 days during which the study was conducted.

Some aspects of reported engagement in protective behaviours were associated with demographic effects, and these are reported in full in electronic supplementary material, Results. In brief, younger adults reported being more likely to attend large events and reported that the virus had caused them to socialize less than older individuals, an effect also observed in females relative to males.

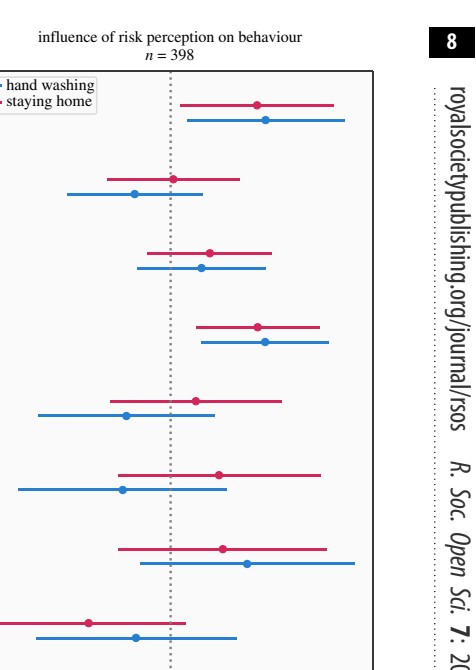

**Figure 3.** Results of linear regression predicting self-reported engagement in hand washing and social distancing (represented by responses to an item regarding staying home) from measures of risk perception, with validation in a subsample of 25% of subjects. (*a*) represents the discovery dataset and (*b*) represents results from the validation dataset. Regression coefficients represent standardized $\beta$ values.

Individuals in the northeast of the country reported engaging in protective behaviours to a greater extent than those in other regions. More striking findings were observed when looking at reported engagement in protective behaviours, where those with college-level education reported engaging in almost all measured protective behaviours to a greater extent than those without a college education.

## 3.3. Influence of risk perception on self-reported protective behaviours

We next investigated the extent to which risk perception was predictive of reported engagement in protective behaviours. We used multiple linear regression to assess the extent to which items assessing risk perception (shown in figure 3) were associated with self-reported engagement in two primary protective behaviours, i.e. hand washing and social distancing, controlling for age. We performed this analysis in a subset consisting of 75% of participants and repeated it in the remaining 25% to ensure the reproducibility of our results. Results were consistent across subsets, and statistics reported here are from the larger dataset (figure 3).

The clearest effect common to both behaviours was a significant effect of perceived *probability* of personally becoming infected (hand washing $\beta = 0.17$, $p < 0.001$, social distancing $\beta = 0.20$, $p < 0.001$, figure 3), while perceived *severity* of illness was not a significant predictor (hand washing $\beta = -0.03$, $p = 0.37$, social distancing $\beta = 0.002$, $p = 0.95$). Perceived impact from global consequences of the pandemic also predicted engagement in both behaviours ($\beta = 0.08$, $p = 0.01$, social distancing $\beta = 0.14$, $p < 0.001$). Notably, the probability of passing the virus on to others and perceived negative effects for another individual who contracted the virus did not significantly predict behaviour (figure 3 and electronic supplementary material, Result). We also performed separate regressions predicting engagement in these protective behaviours based on levels of optimism bias. This revealed significant effects of optimism bias in the initial 75% of the sample, (hand washing $p = 0.001$, social distancing

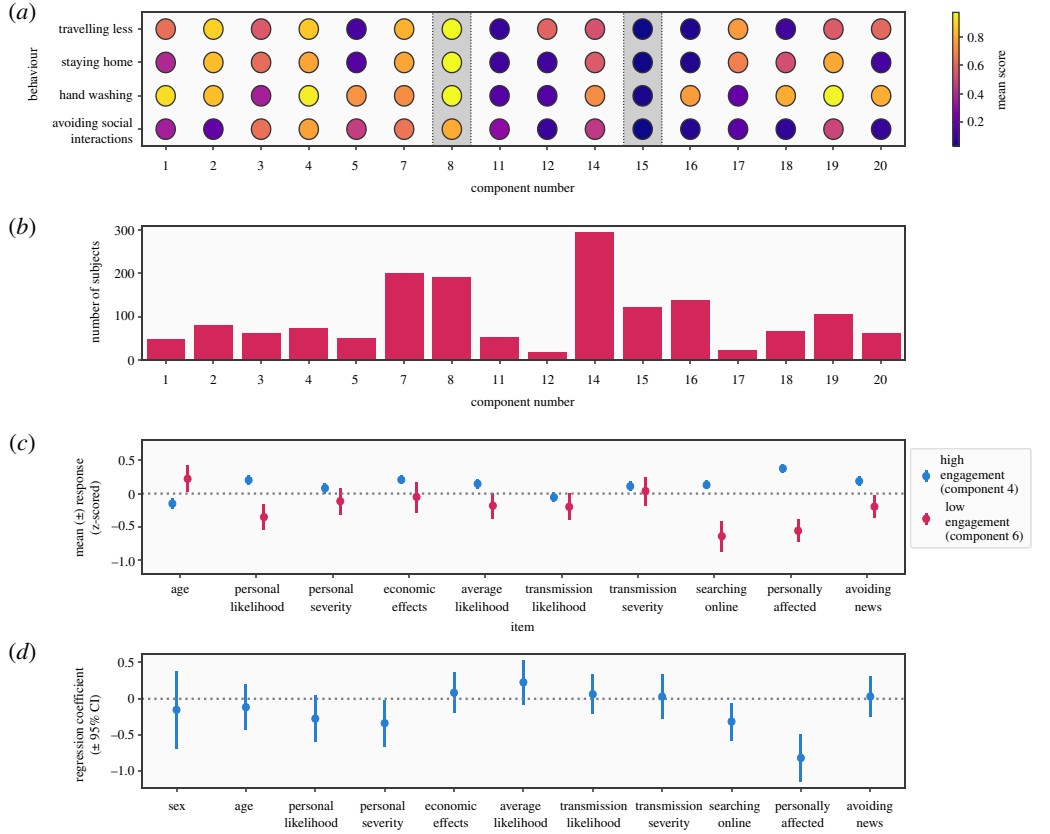

**Figure 4.** Results of Bayesian Gaussian mixture model (GMM) decomposing response distributions for protective behaviour items into clusters. (*a*) Mean scores for each component in the GMM model on the four items used to generate clusters. (*b*) Number of subjects assigned to each group. Four components were rejected due to having negligible weights (less than 0.01). (*c*) Z-scored responses on other questionnaire items for the low engagement and high engagement clusters, demonstrating how they compare to the average individual. (*d*) Results of logistic regression predicting membership of the low engagement group, showing beta coefficients from the model.

$p = 0.001$); however, these were not replicated in the validation sample (hand washing $p = 0.47$, social distancing $p = 0.21$).

These results show that individuals who perceived their probability of becoming infected and believed they would be strongly affected by the broader consequences of the pandemic tended to engage in protective behaviours to a greater extent.

## 3.4. Identification of a subgroup demonstrating low self-reported engagement in protective behaviour

The pattern of responses to questions on protective behaviours was not consistently unimodal (figure 1), suggesting that there are subgroups who behave in qualitatively different ways. To test this, we used a Bayesian Gaussian mixture model (GMM) to decompose the distribution of responses to four primary questions (avoiding social interaction, hand washing, staying home and travelling less) into latent components. Responses on these items were moderately correlated (*r*s ranging from 0.41 to 0.7), suggesting they were measuring related by distinct behaviours. The Bayesian GMM approach allows the number of components to be determined automatically, and this resulted in 16 components being identified. Based on the mean response scores of the components, two components (components 4 and 6) were characterized by high and low reported engagement with protective behaviours, respectively (figure 4*a*).

The model allowed us to assign a probability of each subject being described by each component, which we used to select individuals most likely to belong to the low or high engagement subgroups. There were 135 subjects in the low engagement subgroup and 190 in the high engagement subgroup. We then assessed Z-scored responses to other items to examine how individual subgroups compared

to the group average in terms of perceived risk, information seeking and personal effects of the pandemic (figure 4c). This revealed a broad pattern of below-average perceived risk for both themselves (mean $Z = -0.35$), low engagement with information sources (mean $Z = -0.649$), and low perceived personal effects (mean $Z = -0.56$) in the low engagement subgroup, while the opposite pattern was observed in the high engagement subgroup. The frequency of males and females did not differ significantly from that observed in the group as a whole ($\chi_1^2 = 1.12$, $p = 0.29$).

We next used logistic regression to identify variables that uniquely predicted membership of the low engagement group. This revealed three significant predictors, all of which negatively predicted membership of this group: perceived personal severity of potential infection ($\beta = -0.34$, $p = 0.04$), reported frequency of searching online for information about the virus ($\beta = -0.31$, $p = 0.02$) and feeling personally affected by the pandemic ($\beta = -0.81$, $p < 0.001$). The full results of this analysis are shown in figure 4d and electronic supplementary material, table S11. Together, the results of this analysis identify a subgroup who reported low engagement in a range of protective behaviours, and who tended to feel they had not been personally affected by the pandemic and were not searching for information about the situation.

# 4. Discussion

Understanding how psychological factors influence behaviour in severe, global pandemics such as COVID-19 is key to facilitating disease minimization strategies. Our analyses indicate that, although most individuals are aware of the risk caused by the pandemic to some extent, they typically underestimate their personal risk relative to that of others, an example of optimism bias [12]. In turn, higher perceived personal risk predicts self-reported engagement in protective behaviours, such as hand washing and social distancing, as shown during prior pandemics [9]. Notably, we identified and characterized a non-negligible subset of subjects reporting little-to-no engagement in protective behaviours, who rated overall probability of infection as low and reported being generally disengaged in information seeking and being personally unaffected. Overall, the presence of this subgroup is concerning, given the threat posed by COVID-19 and the beneficial effects of widespread behavioural changes. These results provide insights into the development of psychological and behavioural responses to the early stages of a pandemic. These findings may not necessarily hold during later stages of the pandemic, and we continue to collect additional data to determine how these responses evolve as the pandemic progresses.

Our primary analyses focused on levels of risk perception and self-reported engagement in protective behaviour over time. We found that subjects tended to perceive their personal risk of infection as being lower than the average person in their neighbourhood, state and country, and that reported risk increased between the dates of the study (11 to 16 March 2020). We also found that, on average, subjects reported engaging in many forms of protective behaviour, such as hand washing and social distancing, and reported engagement increased dramatically over the course of the study. Importantly, we verified this pattern of change over time seen in between-subject analyses using repeated testing of a subset of our sample. We additionally investigated predictors of reported engagement in protective behaviours, finding that the clearest predictors were perceived personal risk of infection and being affected by the broader societal consequences of the pandemic. Finally, we sought to identify subgroups of individuals based on their reported engagement in protective behaviours. This analysis identified a subgroup who reported low engagement in all the behaviours studied, and who tended to feel personally unaffected by the pandemic and who were not searching for information about the virus.

One explanation for our results is the optimism bias [12]. This bias is associated with the belief that we are less likely to acquire a disease than others, and has been shown across a variety of diseases including lung cancer [19]. Indeed, those who show the optimism bias are less likely to be vaccinated against disease [20]. Our and other initial data suggest that optimism bias is present with COVID-19 [16]. Our results further show that beliefs about one's own probability of becoming infected predict greater self-reported engagement in protective behaviour, and that behaviour changed over the first week of the COVID-19 pandemic, such that as individuals perceived an increase in personal risk they increasingly engaged in risk-prevention behaviours. Notably, we observed rapid increases in risk perception over a 5-day period, perhaps as a result of public health messages spread through government and the media. This effect was strongest for perceptions of subjects' own risk, diminishing the optimism bias. The speed at which perceptions changed is such that this could have a meaningful effect in terms of reducing disease transmission.

Our results point to candidate targets for intervention in public information campaigns during pandemics. Clear communication of risk could aid the development of accurate risk perception, in turn facilitating engagement in protective behaviours. It would be particularly important to target the subset of individuals who remain disengaged and are not themselves seeking information on the pandemic. Our results suggest that these individuals are not disengaged due to genuine low risk or inability to engage in protective behaviours, being no different in age or education level to the rest of the sample. Instead, membership of this group was predicted by feeling personally unaffected and not seeking information about the pandemic. This suggests the need to expand outreach methods to individuals who do not seek information themselves (e.g. emergency alerts on phones). Furthermore, such disengagement should be considered in epidemiological models used to forecast the effects of behaviour-focused interventions on disease spread. Additionally, education on the beneficial effects of such behaviours for others may improve engagement, particularly in those at low perceived personal risk; it is possible that links between protective behaviour and perceived personal risk minimization are more easily apparent than links with others' health. Finally, the fact that there exists a subgroup who do not engage in protective behaviour implies that these individuals should be accounted for in epidemiological models of these interventions.

These implications should be considered in the context of prior work on the role of fear in motivating beneficial behaviours, however. In particular, one factor that has been frequently highlighted in the literature is self-efficacy. In the presence of a known threat, protective behaviours are most likely to be performed if the individual feels these behaviours will be effective in reducing risk [21] and they will be able to successfully perform the actions necessary to reduce the risk [17]. Similarly, other work emphasizes the importance of engaging in protective behaviours as a response to perceived risk as opposed to fear, and this depends on perceived efficacy [22]. That is, responses to a threat in the context of low perceived efficacy will lead to efforts to reduce fear, such as denial and other coping strategies, while high perceived efficacy will lead to the adoption of behaviours to reduce the risk from the threat itself. One recent study has demonstrated that perceived efficacy predicts self-reported engagement in protective behaviours during the COVID-19 pandemic [23], indicating that this is an area worthy of consideration in efforts to motivate engagement in protective behaviour.

There are limitations to our work that should be considered. First, the median age (30 years) of our sample is relatively young. However, many of our results do not appear to be dependent on age; for example, age was not a significant predictor of hand washing or social distancing. In addition, young people are typically the primary target of efforts to encourage social distancing, having on average larger social networks [24] and therefore a higher probability of engaging in social contact. This is particularly important in the context of COVID-19, where there is evidence that the spread of the virus has been facilitated by the movement of young people with limited to no symptoms [5,6]. Second, our data only reflects views of those in the United States and may not be as applicable to other countries or cultures. It will be important to characterize psychological and behavioural responses across the globe during pandemics in order to recommend and implement the most optimal strategies for effecting behavioural change, which may be culturally specific. It was also not possible to investigate geographical effects thoroughly in this dataset as we only recorded information about location at the state level. It is possible that urbanicity and population density of an individual's local environment could influence their perception of risk during a pandemic. Finally, we also note that the sample was on average at low risk of serious illness due to the pandemic, being majority female and young [25], and therefore the observed optimism bias could be based in being genuinely lower risk than the average person. While we found some evidence that the level of optimism bias was associated with age, this was a weak effect, explaining only 0.5% of the variance in optimism bias. We also note that we did not assess general risk perception levels, and so we cannot determine the extent to which this bias is specific to the pandemic.

One aspect of the relationship between perceived risk and protective behaviour that we were unable to fully account for is the issue of external factors influencing the ability to engage in protective behaviour. The COVID-19 pandemic has affected people differently based on numerous social factors, most notably employment and socioeconomic status, with low-income individuals, often from minority backgrounds, working in environments that necessitate contact with other people [26,27]. Thus, it is possible that variation in perceived risk and reported engagement in protected behaviour here may partially depend on these unmeasured factors, as some individuals are at a genuine high risk of infection and are unable to engage in protective behaviours. However, some of our results paint a more complex picture. When examining differences between college educated and non-college educated individuals, we saw that those with a college-level education both perceived their personal

risk to be higher and reported engaging in protective behaviour to a greater extent. Notably, this included willingness to attend events with large numbers of other people, which is unlikely to be highly dependent on employment in the same way that avoiding social interactions could be. As such, this suggests that individuals with a higher level of education perceive risk as higher and also report engaging in protective behaviours to a greater extent, in a manner that is not purely dependent on employment status.

We also did not address cultural factors that may play a key role in perceptions of risk and engagement in protective behaviours. In other areas where accurate public risk perception and behaviour is required to enable broad beneficial societal change, such as climate change, it has been established that responses can differ dramatically depending on the individual's cultural environment, a phenomenon known as cultural cognition [28]. This prior work has demonstrated that individuals in certain cultures can perceive risk in a way that differs substantially from expert consensus, and this phenomenon may explain the low levels of perceived risk seen here in a subgroup of subjects. Indeed, throughout the pandemic, adherence to public health measures has been found to depend on political affiliation [29], providing some evidence for this hypothesis. Additionally, this study was conducted in the very early stages of the pandemic in the United States, at a time where scientific knowledge about the virus was limited. In the absence of a clear scientific consensus, cultural influences may be exaggerated. Adaptation of behaviour is fundamental to the management of a pandemic on the scale of COVID-19. Our results provide insights into key psychological states and behaviours during a crucial time in the developing situation. Further, our data provide information that may aid authorities to implement precautionary strategies in the event of future pandemics.

Ethics. The study was approved by the Institutional Review Board (IRB) at the California Institute of Technology and subjects provided informed consent.

Data accessibility. All analysis code was written in Python and is available along with data at https://github.com/tobywise/covid19-risk-perception.

Authors' contributions. All authors designed the study. T.W. collected and analysed the data and drafted the manuscript. All authors critically revised the manuscript.

Competing interests. We declare we have no competing interests.

Funding. This work was supported by the US National Institute of Mental Health grant no. 2P50MH094258 and a Chen Institute Award (grant no. P2026052); Merkin Institute grant DM1.COV19R1 and Templeton Foundation grant TWCF0366 (both to D.M.). T.W. is supported by a Wellcome Trust Sir Henry Wellcome Fellowship (grant no. 206460/17/Z). T.D.Z. is supported by the National Science Foundation (grant no. 1911441).

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
