## [Reviewer comments · Royal Society Open Science]

Review History

RSOS-200742.R0 (Original submission)

Review form: Reviewer 1 (Christoph Korn)

Is the manuscript scientifically sound in its present form?

Yes

Are the interpretations and conclusions justified by the results?

Yes

Is the language acceptable?

Yes

Do you have any ethical concerns with this paper?

No

Have you any concerns about statistical analyses in this paper?

No

Recommendation?

Accept with minor revision (please list in comments)

Comments to the Author(s)

Wise and colleagues present a very timely, informative, and well-conducted manuscript on risk perception during the early state of the COVID-19 pandemic in the US. The authors acquired online data from a large sample over multiple time points. Interestingly, they link participants' perceptions of several COVID-19-related risks to several self-reported measures of abiding by best practices of health guidelines. The results are especially important in the current situation but are also generally interesting for understanding the relevance of risk assessments.

The authors' preprint of the data presented here has been very helpful and inspiring for our own study on COVID-19-related optimism, which is still being analyzed. Therefore, I would like to sign this review upfront: My name is Christoph Korn. The two PhD students, Benjamin Kuper-Smith and Lisa Doppelhofer, have helped me write this review. Of course, we will keep everything confidential.

We do not have any major concerns about the study and are very much in favor of publication. We have only rather minor comments that mainly aim at making the manuscript clearer. We would also take the opportunity to ask a few questions about possible additional measures that the authors might have assessed.

1) Timing of data acquisition:

a. It was not quite clear to us, how data for T1 was combined. Was this from March 11 to 15? Are there any changes apparent in between-participants analyses? These would probably be similar to the reported within-participant differences.

b. In Fig 1, it would be nice to see the distributions of the (within-participant) changes; e.g., in some line graphs.

2) Ordering of questions and presentation of figures and tables:

a. In the text, the authors mention their "four primary questions" for the Gaussian Mixture Models but for the regressions they only report two, i.e., hand washing and staying at home. This could be streamlined and better motivated, which would also help to organize the figures and tables.

b. This would also allow the readers to dissociate which analyses were determined a priori and which were more exploratory. Ideally, this would be mentioned a bit more clearly.

c. How are the four (or two) primary self-reported protective behaviors related to each other? Fig. 3 suggests that hand washing and staying at home are substantially correlated.

d. Overall, it was not quite clear how the ordering of the questions in the figures and tables was chosen. It would be nice if these could all be consistent.

e. If subplots had the same scale, comparisons would be easier; e.g., Fig 3A & B.

f. Some text parts in the figures, e.g., in Fig. 1, are quite small.

g. It would be good to have precise p-values in the figures.

3) Focus on US & questions about geography:

a. Overall, it is clear that all participants are from the US. In some cases, the wording might be a bit confusing for non-US-residents, e.g., "all 50 states" could be "all 50 US states," etc. Sometimes "government" could be "US government." CDC could be very briefly explained.

b. It takes some search in the supplementary to find out how participants were split according to geography. This could be easily described more clearly in the main text. The authors just state that they tested for geography (e.g., p. 8/14), but don't provide any details about what test they use or what the result were.

c. Did the authors test for any effects of population density (e.g., city versus countryside, etc.) or of disease status in the different states? Maybe some participants had a low risk perception because they live in the countryside and/or because their state had a low infection rate.

4) Average person:

a. Did the question about an average person control for variables that are relevant for COVID-19 (or at least seemed like it at the time; e.g., age, gender, location)?

b. Did the authors look at the differences between perceived risks for self and for the average person in the state? This would be an index of comparative optimism, which could be more indicative of some self-reported protective behaviors than "pure" risk perceptions.

- c. Btw. do current estimates still suggest that 80% of the population may contract COVID-19 (bottom p. 5)?
- d. It would be nice to read a bit more of a discussion about to what extent the differences between self and average might make sense e.g., the sample was 55% female and largely young, so it might make sense that they don't fear the virus as much, given that they are pretty much the furthest away from being a risk population as possible (this could be briefly done in the discussion)
- e. Did the authors look into education level? It seems for example that white-collar workers can more easily reduce physical contact than blue-collar works.
- 5) Subgroups/ Gaussian Mixture Model: We have no expertise in this type of modelling so we cannot adequately assess the rigor of this part of the analysis.
- a. Could the authors characterize the clusters a bit more? They provide almost no information (or even discussion about) to what extent a low level of engagement might be justified. Maybe the people who engage little are absolutely justified in doing so and it's just a rational belief.
- b. How many people are in those clusters? Is it a lot of people or just very few? This makes a difference because you'd expect a certain amount of people in each cluster based on random effects, but it would be interesting if e.g., the low-engagement cluster is much larger than would be expected.
- c. Are the number of subgroups pre-specified in the analyses?
- 6) Additional data:
- a. Did the authors collect data if restrictions were perceived as forced versus voluntary?
- b. Did they assess risks for events that are unrelated to COVID-19 to get some baseline for risk perception?
- c. Did the authors assess any links between optimism bias and increased coping mechanism/better mental health?
- d. It seems quite likely that the authors continue to collect data. If so, it would be nice if they could hint at this in the discussion. Readers could thus look out for new publications.
- 7) Some further points
- a. Probability & likelihood are not the same statistically. Please try to be as precise as possible.
- b. The authors can only analyze self-reported behavior. This is mentioned and it's also obvious from the study design but it could be highlighted a bit more, for example in the title: "self-reported protective behavior"
- c. Please add a reference for claim about when which restrictions were made in US relative to data collection (p. 4/14 bottom)

Review form: Reviewer 2

Is the manuscript scientifically sound in its present form?

Yes

Are the interpretations and conclusions justified by the results?

No

Is the language acceptable?

Yes

Do you have any ethical concerns with this paper?

No

Have you any concerns about statistical analyses in this paper?

Yes

Recommendation?

Accept with minor revision (please list in comments)

Comments to the Author(s)

- 1) It would be better to place the figure 1 in the section of results so that the descriptions in the text and figure would align.
- 2) In the study design, can you justify the combined cross sectional and longitudinal study along with its uses in the study?
- 3) Please review the results with more descriptions and explanations.
- 4) The discussion should focus on the results with evidences.

Review form: Reviewer 3**Is the manuscript scientifically sound in its present form?**

Yes

Are the interpretations and conclusions justified by the results?

Yes

Is the language acceptable?

Yes

Do you have any ethical concerns with this paper?

No

Have you any concerns about statistical analyses in this paper?

No

Recommendation?

Accept with minor revision (please list in comments)

Comments to the Author(s)

This manuscript is concerned with perceptions of risk to the self and to others relating to the current COVID-19 global pandemic and associated protective actions.

Overall I think this is an exemplary, thoughtful and timely piece of applied research. A couple of the figures are a little bit busy but they are also good at clearly presenting a lot of information so I think it's a worthwhile tradeoff to make. The one place I think could use a little more attention would be the discussion section, there are a couple of places where some additional theory could help to bolster or provide potential explanations for some of the results as well as add nuance to some of the recommendations. With minor revisions of this kind I would recommend publication of this manuscript.

Methods:

One thing that came to mind was whether there were any problematic intercorrelations between the risk perception items, was there any evidence of high intercorrelations that might have masked some of the results in the linear regressions?

Results and Discussion:

Another thing I think is very interesting is looking at the disengaged group. While this group doesn't appear to differ by gender (which might be expected if it's the result of something like the "white male effect" as in Finucane et al, 2000), I think there might be something to the idea that

this group is driven by other cultural or socio-political motivations. While obviously you do not necessarily have to data to conclusively rule out or support something like a cultural cognition argument, it might be worth mentioning some of that literature (for instance the Kahan et al, 2011 below) in the discussion. It seems like there are some similar points to be made about scientific consensus with regards to the pandemic that are routinely made with respect to scientific consensus on climate change. I'm having a little trouble following the Bayesian analysis myself but I would be interested to see if elevated concerns with respect to economic impacts from the global response to the virus might characterize one group over another and whether a cultural argument could be made for such differences. To be clear, I don't necessarily think that this requires a change to the manuscript, but I think some of these issues might help to add nuance to the discussion, particularly with reference to theory beyond the optimism bias argument.

Another potential source of nuance in the discussion might come from fear appeal literature such as that from Protection Motivation Theory (Rogers, 1975; Maddux & Rogers, 1983) and the Extended Parallel Process model (Witte, 1992), both of which suggest that substantive change in cognitions related to protective action in response to fear appeals (in this case information from government and media sources concerning covid and the appropriate responses) will be the result of both high risk perception and high self- and response efficacy (i.e. "There's something I can do" and "If I do it, it will work"). In particular, the extended parallel process model would suggest that refraining from protective action might be the result of one of two potential processes, either because risk perception is initially low (resulting in no action), or risk perception is high but one or more efficacy variable is low leading to a motivation to control fear as opposed to controlling danger which can lead to a reduction in risk perception (that is, if I am scared but there's nothing I think I can do that will help, I downplay the risk to make myself more comfortable). These may also be worth considering when thinking about how to engage the disengaged group beyond additional education efforts in the discussion.

Finucane, M. L., Slovic, P., Mertz, C. K., Flynn, J., & Satterfield, T. A. (2000). Gender, race, and perceived risk: The 'white male' effect. *Health, risk & society*, 2(2), 159-172.

Kahan, D. M., Jenkins-Smith, H., & Braman, D. (2011). Cultural cognition of scientific consensus. *Journal of risk research*, 14(2), 147-174.

Maddux, J. E., & Rogers, R. W. (1983). Protection motivation and self-efficacy: A revised theory of fear appeals and attitude change. *Journal of experimental social psychology*, 19(5), 469-479.

Rogers, R. W. (1975). A protection motivation theory of fear appeals and attitude change. *The journal of psychology*, 91(1), 93-114.

Witte, K. (1992). Putting the fear back into fear appeals: The extended parallel process model. *Communications Monographs*, 59(4), 329-349.

Minor General Notes

Page 2 Line 51: I believe this sentence should read "we are not aware of any data..." or "we are aware of no data..."

Page 3 Line 50: "survey" should probably be pluralized

Page 5 Line 31: "the a multivariate distribution" should use either "the" or "a" but not both

Page 8 Line 58: I believe this should be "Bayesian Gaussian mixture model"

Decision letter (RSOS-200742.R0)

Dear Dr Wise

On behalf of the Editors, I am pleased to inform you that your Manuscript RSOS-200742 entitled "Changes in risk perception and protective behavior during the first week of the COVID-19 pandemic in the United States" has been accepted for publication in Royal Society Open Science subject to minor revision in accordance with the referee suggestions. Please find the referees' comments at the end of this email.

The reviewers and handling editors have recommended publication, but also suggest some minor revisions to your manuscript. Therefore, I invite you to respond to the comments and revise your manuscript.

- Ethics statement

- Data accessibility

<http://datadryad.org/submit?journalID=RSOS&manu=RSOS-200742>

- Competing interests

- Authors' contributions

- Acknowledgements

- Funding statement

Because the schedule for publication is very tight, it is a condition of publication that you submit the revised version of your manuscript before 29-Jul-2020. Please note that the revision deadline will expire at 00.00am on this date. If you do not think you will be able to meet this date please let me know immediately.

- 1) A text file of the manuscript (tex, txt, rtf, docx or doc), references, tables (including captions) and figure captions. Do not upload a PDF as your "Main Document";
- 2) A separate electronic file of each figure (EPS or print-quality PDF preferred (either format should be produced directly from original creation package), or original software format);
- 3) Included a 100 word media summary of your paper when requested at submission. Please ensure you have entered correct contact details (email, institution and telephone) in your user account;
- 4) Included the raw data to support the claims made in your paper. You can either include your data as electronic supplementary material or upload to a repository and include the relevant doi within your manuscript. Make sure it is clear in your data accessibility statement how the data can be accessed;
- 5) All supplementary materials accompanying an accepted article will be treated as in their final form. Note that the Royal Society will neither edit nor typeset supplementary material and it will

be hosted as provided. Please ensure that the supplementary material includes the paper details where possible (authors, article title, journal name).

If your manuscript is newly submitted and subsequently accepted for publication, you will be asked to pay the article processing charge, unless you request a waiver and this is approved by Royal Society Publishing. You can find out more about the charges at <https://royalsocietypublishing.org/rsos/charges>. Should you have any queries, please contact openscience@royalsociety.org.

on behalf of Dr Christina Demski (Associate Editor) and Essi Viding (Subject Editor)
openscience@royalsociety.org

Associate Editor Comments to Author (Dr Christina Demski):
Comments to the Author:

The reviewers agreed that the manuscript provides a timely and scientifically sound analysis and have recommended it for publication. However, they would like some revisions and additional discussions before it can be published. The discussion section in particular is highlighted to need some improvement.

Reviewer comments to Author:
Reviewer: 1

Comments to the Author(s)

Wise and colleagues present a very timely, informative, and well-conducted manuscript on risk perception during the early state of the COVID-19 pandemic in the US. The authors acquired online data from a large sample over multiple time points. Interestingly, they link participants' perceptions of several COVID-19-related risks to several self-reported measures of abiding by best practices of health guidelines. The results are especially important in the current situation but are also generally interesting for understanding the relevance of risk assessments.

The authors' preprint of the data presented here has been very helpful and inspiring for our own study on COVID-19-related optimism, which is still being analyzed. Therefore, I would like to sign this review upfront: My name is Christoph Korn. The two PhD students, Benjamin Kuper-Smith and Lisa Doppelhofer, have helped me write this review. Of course, we will keep everything confidential.

We do not have any major concerns about the study and are very much in favor of publication. We have only rather minor comments that mainly aim at making the manuscript clearer. We would also take the opportunity to ask a few questions about possible additional measures that the authors might have assessed.

1) Timing of data acquisition:

a. It was not quite clear to us, how data for T1 was combined. Was this from March 11 to 15? Are there any changes apparent in between-participants analyses? These would probably be similar to the reported within-participant differences.

b. In Fig 1, it would be nice to see the distributions of the (within-participant) changes; e.g., in some line graphs.

2) Ordering of questions and presentation of figures and tables:

a. In the text, the authors mention their "four primary questions" for the Gaussian Mixture Models but for the regressions they only report two, i.e., hand washing and staying at home. This could be streamlined and better motivated, which would also help to organize the figures and tables.

b. This would also allow the readers to dissociate which analyses were determined a priori and which were more exploratory. Ideally, this would be mentioned a bit more clearly.

c. How are the four (or two) primary self-reported protective behaviors related to each other? Fig. 3 suggests that hand washing and staying at home are substantially correlated.

d. Overall, it was not quite clear how the ordering of the questions in the figures and tables was chosen. It would be nice if these could all be consistent.

e. If subplots had the same scale, comparisons would be easier; e.g., Fig 3A & B.

f. Some text parts in the figures, e.g., in Fig. 1, are quite small.

g. It would be good to have precise p-values in the figures.

3) Focus on US & questions about geography:

a. Overall, it is clear that all participants are from the US. In some cases, the wording might be a bit confusing for non-US-residents, e.g., "all 50 states" could be "all 50 US states," etc. Sometimes "government" could be "US government." CDC could be very briefly explained.

b. It takes some search in the supplementary to find out how participants were split according to geography. This could be easily described more clearly in the main text. The authors just state that they tested for geography (e.g., p. 8/14), but don't provide any details about what test they use or what the result were.

c. Did the authors test for any effects of population density (e.g., city versus countryside, etc.) or of disease status in the different states? Maybe some participants had a low risk perception because they live in the countryside and/or because their state had a low infection rate.

4) Average person:

a. Did the question about an average person control for variables that are relevant for COVID-19 (or at least seemed like it at the time; e.g., age, gender, location)?

b. Did the authors look at the differences between perceived risks for self and for the average person in the state? This would be an index of comparative optimism, which could be more indicative of some self-reported protective behaviors than "pure" risk perceptions.

c. Btw. do current estimates still suggest that 80% of the population may contract COVID-19 (bottom p. 5)?

d. It would be nice to read a bit more of a discussion about to what extent the differences between self and average might make sense e.g., the sample was 55% female and largely young, so it might make sense that they don't fear the virus as much, given that they are pretty much the furthest away from being a risk population as possible (this could be briefly done in the discussion)

e. Did the authors look into education level? It seems for example that white-collar workers can more easily reduce physical contact than blue-collar workers.

- 5) Subgroups/ Gaussian Mixture Model: We have no expertise in this type of modelling so we cannot adequately assess the rigor of this part of the analysis.
- Could the authors characterize the clusters a bit more? They provide almost no information (or even discussion about) to what extent a low level of engagement might be justified. Maybe the people who engage little are absolutely justified in doing so and it's just a rational belief.
 - How many people are in those clusters? Is it a lot of people or just very few? This makes a difference because you'd expect a certain amount of people in each cluster based on random effects, but it would be interesting if e.g., the low-engagement cluster is much larger than would be expected.
 - Are the number of subgroups pre-specified in the analyses?
- 6) Additional data:
- Did the authors collect data if restrictions were perceived as forced versus voluntary?
 - Did they assess risks for events that are unrelated to COVID-19 to get some baseline for risk perception?
 - Did the authors assess any links between optimism bias and increased coping mechanism/better mental health?
 - It seems quite likely that the authors continue to collect data. If so, it would be nice if they could hint at this in the discussion. Readers could thus look out for new publications.
- 7) Some further points
- Probability & likelihood are not the same statistically. Please try to be as precise as possible.
 - The authors can only analyze self-reported behavior. This is mentioned and it's also obvious from the study design but it could be highlighted a bit more, for example in the title: "self-reported protective behavior"
 - Please add a reference for claim about when which restrictions were made in US relative to data collection (p. 4/14 bottom)

Reviewer: 2

Comments to the Author(s)

- It would be better to place the figure 1 in the section of results so that the descriptions in the text and figure would align.
- In the study design, can you justify the combined cross sectional and longitudinal study along with its uses in the study?
- Please review the results with more descriptions and explanations.
- The discussion should focus on the results with evidences.

Reviewer: 3

Comments to the Author(s)

This manuscript is concerned with perceptions of risk to the self and to others relating to the current COVID-19 global pandemic and associated protective actions.

Overall I think this is an exemplary, thoughtful and timely piece of applied research. A couple of the figures are a little bit busy but they are also good at clearly presenting a lot of information so I think it's a worthwhile tradeoff to make. The one place I think could use a little more attention would be the discussion section, there are a couple of places where some additional theory could help to bolster or provide potential explanations for some of the results as well as add nuance to some of the recommendations. With minor revisions of this kind I would recommend publication of this manuscript.

Methods:

One thing that came to mind was whether there were any problematic intercorrelations between the risk perception items, was there any evidence of high intercorrelations that might have masked some of the results in the linear regressions?

Results and Discussion:

Another thing I think is very interesting is looking at the disengaged group. While this group doesn't appear to differ by gender (which might be expected if it's the result of something like the "white male effect" as in Finucane et al, 2000), I think there might be something to the idea that this group is driven by other cultural or socio-political motivations. While obviously you do not necessarily have to data to conclusively rule out or support something like a cultural cognition argument, it might be worth mentioning some of that literature (for instance the Kahan et al, 2011 below) in the discussion. It seems like there are some similar points to be made about scientific consensus with regards to the pandemic that are routinely made with respect to scientific consensus on climate change. I'm having a little trouble following the Bayesian analysis myself but I would be interested to see if elevated concerns with respect to economic impacts from the global response to the virus might characterize one group over another and whether a cultural argument could be made for such differences. To be clear, I don't necessarily think that this requires a change to the manuscript, but I think some of these issues might help to add nuance to the discussion, particularly with reference to theory beyond the optimism bias argument.

Another potential source of nuance in the discussion might come from fear appeal literature such as that from Protection Motivation Theory (Rogers, 1975; Maddux & Rogers, 1983) and the Extended Parallel Process model (Witte, 1992), both of which suggest that substantive change in cognitions related to protective action in response to fear appeals (in this case information from government and media sources concerning covid and the appropriate responses) will be the result of both high risk perception and high self- and response efficacy (i.e. "There's something I can do" and "If I do it, it will work"). In particular, the extended parallel process model would suggest that refraining from protective action might be the result of one of two potential processes, either because risk perception is initially low (resulting in no action), or risk perception is high but one or more efficacy variable is low leading to a motivation to control fear as opposed to controlling danger which can lead to a reduction in risk perception (that is, if I am scared but there's nothing I think I can do that will help, I downplay the risk to make myself more comfortable). These may also be worth considering when thinking about how to engage the disengaged group beyond additional education efforts in the discussion.

Finucane, M. L., Slovic, P., Mertz, C. K., Flynn, J., & Satterfield, T. A. (2000). Gender, race, and perceived risk: The 'white male' effect. *Health, risk & society*, 2(2), 159-172.

Kahan, D. M., Jenkins-Smith, H., & Braman, D. (2011). Cultural cognition of scientific consensus. *Journal of risk research*, 14(2), 147-174.

Maddux, J. E., & Rogers, R. W. (1983). Protection motivation and self-efficacy: A revised theory of fear appeals and attitude change. *Journal of experimental social psychology*, 19(5), 469-479.

Rogers, R. W. (1975). A protection motivation theory of fear appeals and attitude change. *The journal of psychology*, 91(1), 93-114.

Witte, K. (1992). Putting the fear back into fear appeals: The extended parallel process model. *Communications Monographs*, 59(4), 329-349.

Minor General Notes

Page 2 Line 51: I believe this sentence should read "we are not aware of any data..." or "we are aware of no data..."

Page 3 Line 50: "survey" should probably be pluralized

Page 5 Line 31: "the a multivariate distribution" should use either "the" or "a" but not both

Page 8 Line 58: I believe this should be "Bayesian Gaussian mixture model"

Author's Response to Decision Letter for (RSOS-200742.R0)

See Appendices A & B.

RSOS-200742.R1 (Revision)

Review form: Reviewer 2

Is the manuscript scientifically sound in its present form?

Yes

Are the interpretations and conclusions justified by the results?

Yes

Is the language acceptable?

Yes

Do you have any ethical concerns with this paper?

Yes

Have you any concerns about statistical analyses in this paper?

No

Recommendation?

Accept as is

Comments to the Author(s)

All the previous comments have been addressed. Seems ok.

Review form: Reviewer 3

Is the manuscript scientifically sound in its present form?

Yes

Are the interpretations and conclusions justified by the results?

Yes

Is the language acceptable?

Yes

Do you have any ethical concerns with this paper?

No

Have you any concerns about statistical analyses in this paper?

No

Recommendation?

Accept as is

Comments to the Author(s)

Thank you for your extensive attention to addressing all of the reviewers comments! I think you have a very interesting manuscript here!

Decision letter (RSOS-200742.R1)

Dear Dr Wise,

It is a pleasure to accept your manuscript entitled "Changes in risk perception and self-reported protective behavior during the first week of the COVID-19 pandemic in the United States" in its current form for publication in Royal Society Open Science. The comments of the reviewers who reviewed your manuscript are included at the foot of this letter.

COVID-19 rapid publication process:

We are taking steps to expedite the publication of research relevant to the pandemic. If you wish, you can opt to have your paper published as soon as it is ready, rather than waiting for it to be published the scheduled Wednesday.

This means your paper will not be included in the weekly media round-up which the Society sends to journalists ahead of publication. However, it will still appear in the COVID-19 Publishing Collection which journalists will be directed to each week (<https://royalsocietypublishing.org/topic/special-collections/novel-coronavirus-outbreak>).

If you wish to have your paper considered for immediate publication, or to discuss further, please notify openscience_proofs@royalsociety.org and press@royalsociety.org when you respond to this email.

on behalf of Dr Christina Demski (Associate Editor) and Essi Viding (Subject Editor)
openscience@royalsociety.org

Associate Editor Comments to Author (Dr Christina Demski):

Two reviewers have now looked at the revisions that were made and have indicated that the manuscript is ready for acceptance. Thank you for addressing all the comments in such a detailed manner.

Reviewer comments to Author:

Reviewer: 2

Comments to the Author(s)

All the previous comments have been addressed. Seems ok.

Reviewer: 3

Comments to the Author(s)

Thank you for your extensive attention to addressing all of the reviewers comments! I think you have a very interesting manuscript here!

Appendix A

Humanities and Social Sciences
Computation and Neural Systems

1200 E. California Blvd., MC 0-00
Pasadena, CA 91125
(626) 395-4028
tobywise@caltech.edu

30th July, 2020

Dear Dr Demski,

Thank you for your consideration of our manuscript and for the opportunity to resubmit.

We have revised the paper in line with the reviewers' comments, and we believe the manuscript is greatly improved thanks to their input. Changes are highlighted in the manuscript and provided in the response to the reviewers.

We look forward to your response.

Sincerely,

Toby Wise, PhD
Social, Affective and Ecological Neuroscience Laboratory,
California Institute of Technology (Caltech)
Pasadena, California, 91125,
Phone: +1 626 714 8913, Email: tobywise@caltech.edu

Appendix B

Associate Editor Comments to Author (Dr Christina Demski):

Comments to the Author:

The reviewers agreed that the manuscript provides a timely and scientifically sound analysis and have recommended it for publication. However, they would like some revisions and additional discussions before it can be published. The discussion section in particular is highlighted to need some improvement.

Thank you for your consideration of our manuscript and the positive comments. We hope we have been able to address the reviewers' comments satisfactorily.

Reviewer comments to Author:

Reviewer: 1

Comments to the Author(s)

Wise and colleagues present a very timely, informative, and well-conducted manuscript on risk perception during the early state of the COVID-19 pandemic in the US. The authors acquired online data from a large sample over multiple time points. Interestingly, they link participants' perceptions of several COVID-19-related risks to several self-reported measures of abiding by best practices of health guidelines. The results are especially important in the current situation but are also generally interesting for understanding the relevance of risk assessments.

The authors' preprint of the data presented here has been very helpful and inspiring for our own study on COVID-19-related optimism, which is still being analyzed. Therefore, I would like to sign this review upfront: My name is Christoph Korn. The two PhD students, Benjamin Kuper-Smith and Lisa Doppelhofer, have helped me write this review. Of course, we will keep everything confidential.

We do not have any major concerns about the study and are very much in favor of publication. We have only rather minor comments that mainly aim at making the manuscript clearer. We would also take the opportunity to ask a few questions about possible additional measures that the authors might have assessed.

Thank you to you and your PhD students for your valuable comments. We are also glad that our paper has been helpful for your own work, and were likewise inspired by your recent preprint on the topic.

Your suggestions have substantially improved the manuscript and we hope we have been able to address your comments satisfactorily.

1) Timing of data acquisition:

a. It was not quite clear to us, how data for T1 was combined. Was this from March 11 to 15? Are there any changes apparent in between-participants analyses? These would probably be similar to the reported within-participant differences.

Thank you for noting this, we agree that it was not sufficiently clear in the original manuscript. For T1, we used just the data from March 11th, as these were the subjects who were followed up for T2. We have added the following to the methods explain this:

"For longitudinal analyses, we compared data from this follow-up time point (referred to as T2) with data from these same subjects collected on 3/11/2020 (referred to as T1)."

Regarding between-subject changes over time, we did find increases in perceived risk and engagement in protective behavior, consistent with the effects in the within-subject analysis. This was reported in the original manuscript, but to increase clarity we have rephrased it as follows:

"In between subjects analyses, perceived likelihood of infection differed across samples tested on different days, demonstrating a higher rate in subjects tested on later dates ($F(6, 1579) = 6.48, p <$

.001, $\eta_p^2 = 0.024$, Figure 2A). To confirm that represented a change over time, we examined increases in perceived likelihood within-subjects in a subsample followed up after 5 days, finding that this did indeed increase over time ($F(1, 374) = 69.19, p < .001, \eta_p^2 = 0.16$, Figure 2B). In this within-subjects analysis there was an interaction between time and the person being rated ($F(3, 1122) = 7.56, p < .001, \eta_p^2 = 0.02$),”

“In between-subjects analyses, we also saw markedly lower likelihood ratings over time in separate samples collected across multiple days ($F(6, 1579) = 22.84, p < .001, \eta_p^2 = 0.08$, Figure 2C). Congruently, a decrease over time emerged in our within-subject analysis ($F(1, 374) = 279.02, p < .001, \eta_p^2 = 0.43$, Figure 2D)”

We have also added a section to the methods to describe these analyses in more detail:

“We tested for changes over time in two ways: First, we used mixed ANOVAs to test for differences in responses across samples collected on different dates. For analyses focusing on risk perception, the dependent variable was perceived likelihood of infection. The within subject factor was the subject of the rating (self, average person in the neighborhood, average person in the state, average person in the country), and the between subject factor was the date of testing. For analyses focusing on behavior, our dependent variable was likelihood of attending an event with a specific number of other people (10, 50, 100, 500 or 1000), with the within subject factor being the number of people and the between subject factor being the date of testing.

Second, we used repeated ANOVAs in our longitudinal dataset to test for within-subject changes over time. Here, the factors were the same as above, however date of testing was a within-subject factor in this case.”

b. In Fig 1, it would be nice to see the distributions of the (within-participant) changes; e.g., in some line graphs.

Thank you for this suggestion, this is indeed a helpful way to visualize patterns of change for every subject. We have added figures for all within-subject effects shown in Figure 1 to supplementary material (Figure S9 and S10) and we have signposted these figures in the legend for Figure 1 in the main manuscript.

2) Ordering of questions and presentation of figures and tables:

a. In the text, the authors mention their “four primary questions” for the Gaussian Mixture Models but for the regressions they only report two, i.e., hand washing and staying at home. This could be streamlined and better motivated, which would also help to organize the figures and tables.

We agree that this could be streamlined and motivated more clearly. Our reason for focusing on two items for the regressions was that we wanted to minimize the number of analyses we conducted. In contrast, including more measures was not an issue for the Gaussian mixture models, as this took a multivariate exploratory approach, and a greater number of variables provides richer data for the model. We have added the following to the methods section to clarify the reason for selecting these variables:

"While we did ask questions relating to a wider range of protective behaviors, we elected to limit our analyses to these two measures to reduce the complexity of our analyses and avoid testing large numbers of variables for potential associations. These two items were chosen as they represented behaviors that were known to help reduce the spread of the virus at the time."

"For this data-driven clustering analysis, we used four measures of engagement in protective behaviours (avoiding social interaction, hand-washing, staying home, and travelling less) as using multiple indicator variables provides more information from which to identify clusters."

For completeness, we have also run regressions using the additional two measures and added these to supplementary material. These show broadly similar results to the analyses with the original two measures, and are reported in Figure S13 and Tables S9 and S10.

b. This would also allow the readers to dissociate which analyses were determined a priori and which were more exploratory. Ideally, this would be mentioned a bit more clearly.

Thank you for this suggestion. We had clear hypotheses regarding increases in risk perception and protective behavior, along with the optimism bias in perceived risk. Our analyses focusing on predictors of engagement in protective behavior were exploratory, although we followed these with confirmatory analyses in an independent group of subjects to ensure the results were robust. Finally, the identification of subgroups with Gaussian Mixture Models was purely exploratory. We have added the following to the manuscript to describe this. For the analyses of change over time:

"All of the analyses related to changes over time were determined a priori, and we hypothesized that perceived risk would increase over time, while willingness to attend events with other people would decrease over time. We also expected subjects to rate infection probability as being higher for others than for themselves and expected to see lower willingness to attend events with higher numbers of people."

For the analyses of predictors of protective behavior:

"These analyses were exploratory, and so we first performed this analysis in a discovery dataset consisting of 75% of subjects' data, randomly selected. We then ran the same analysis in the remaining 25% to ensure replicability of our results."

And for the subgroups analysis:

"This analysis was entirely exploratory, we did not have a priori hypotheses regarding the subgroups and instead allowed these to be derived from the data."

c. How are the four (or two) primary self-reported protective behaviors related to each other? Fig. 3 suggests that hand washing and staying at home are substantially correlated.

Thank you for this suggestion. The measures were indeed correlated to a moderate extent (with r s ranging from .41 to .7). We have added inter-item correlations to supplementary material (Figure S12) and have mentioned this in the results as follows:

"Responses on these items were moderately correlated (rs ranging from .41 to .7), suggesting they were measuring related by distinct behaviors."

d. Overall, it was not quite clear how the ordering of the questions in the figures and tables was chosen. It would be nice if these could all be consistent.

Thank you for this suggestion, we appreciate that the order of questions was not as consistent as it could have been. We have modified the relevant figures and tables to ensure that the order of items is consistent.

e. If subplots had the same scale, comparisons would be easier; e.g., Fig 3A & B.

We have rescaled the axes on Figures 3A and 3B (and other figures showing coefficients from different regression models) to facilitate comparisons between the two.

f. Some text parts in the figures, e.g., in Fig. 1, are quite small.

Thank you for noting this. We have tried to increase the text as much as possible, although given the number of figures there is unfortunately a limit to how large we can make the text. In general, we have consciously chosen to present as much data as possible here, rather than selectively reporting specific items. We appreciate that this does mean the figures may not be as clear as they could be, but we believe this is a sacrifice worth making.

g. It would be good to have precise p-values in the figures.

Thank you for this suggestion. We have now replaced indications of significance in the relevant figures with exact p-values.

3) Focus on US & questions about geography:

a. Overall, it is clear that all participants are from the US. In some cases, the wording might be a bit confusing for non-US-residents, e.g., "all 50 states" could be "all 50 US states," etc. Sometimes "government" could be "US government." CDC could be very briefly explained.

Thank you, we have made changes to the US-specific terms throughout (we have not listed these here as they are numerous and small), and have included a brief definition of the CDC as follows:

"Notably, testing occurred before and after the recommendation by the CDC (Centers for Disease Control and Prevention, the United States public health institute) of avoiding gatherings of 50+ people"

b. It takes some search in the supplementary to find out how participants were split according to geography. This could be easily described more clearly in the main text. The authors just state that they tested for geography (e.g., p. 8/14), but don't provide any details about what test they use or what the result were.

Thank you for this suggestion. We have moved the description from the supplementary material to the methods section, and we hope that this provides more clarity regarding this analysis:

"We also explored effects of age, sex, and location on risk perception and engagement in protective behaviours. To examine effects of age, we split subjects into three groups based on age (18-26, 26-35, 35-80) using a tertile split. For analyses involving location, we assigned each individual to a census-defined region according to their state (<https://www.census.gov/geographies/reference-maps/2010/geo/2010-census-regions-and-divisions-of-the-united-states.html>), resulting in each subject being allocated to one of the following regions: South, West, Midwest, or North East. Only subjects where location data was available were included in this analysis.

We used ANOVAs to examine effects of group on responses in the case of age and location, and t-tests in the case of sex, and education level, with p values corrected for multiple comparisons using false discovery rate (FDR) correction. Full results for these analyses are provided in Tables S1-S8, and Figures S3-S10."

c. Did the authors test for any effects of population density (e.g., city versus countryside, etc.) or of disease status in the different states? Maybe some participants had a low risk perception because they live in the countryside and/or because their state had a low infection rate.

This is an interesting question, and it is something we considered investigating. However, we unfortunately only had ethical approval to collect information about location at the state level (as opposed to country or zip-code level), which does not necessarily tell us much about population density (some states have low overall population density, but densely populated urban areas for example). In more recent data collection using a subset of this sample, we have collected information about the nature of area in which they live however, and we are aiming to look at associations with risk perception in future studies. We have mentioned this issue in the discussion:

"It was also not possible to investigate geographical effects thoroughly in this dataset as we only recorded information about location at the state level. It is possible that urbanicity and population density of an individual's local environment could influence their perception of risk during a pandemic."

4) Average person:

a. Did the question about an average person control for variables that are relevant for COVID-19 (or at least seemed like it at the time; e.g., age, gender, location)?

Thank you for this suggestion, this is an important question. Our analyses did not control for these variables, but it is important to consider whether perhaps the optimism bias we see is dependent on genuine risk relative to the average person. To test this directly, we have performed an additional analysis predicting optimism bias (difference in reported infection probability for self versus state average) from age, sex, and location. This showed only a weak effect of age. We have reported these results as follows in the text:

"It is possible that the observed optimism bias here may reflect the relatively low risk level of the sample, potentially being at genuinely lower risk than the average person in their state or country. To

assess the influence of these factors, we used an ANOVA predicting optimism bias (the difference between perceived risk for the self and that for the average person in the state) from age, sex, and location. This revealed a significant, albeit weak, negative effect of age ($F(1, 873) = 5.87, p = .016$). Effects of sex and location were not significant ($ps = .77$ and $.65$ respectively). We followed this by using a linear regression predicting optimism bias scores from age alone to determine the proportion of variance explained by age. This resulted in an R^2 of 0.005, indicating that only 0.5% of the variance in optimism bias scores was explained by age."

In addition, we have run an additional analysis predicting group status (i.e. the low engagement subgroup versus the rest of the sample) from the various variables of interest using logistic regression, to identify those that uniquely predict group membership controlling for all other variables. We elected to exclude location from this analysis because this data was not collected for approximately one third of the sample. We have described these analyses more thoroughly in the methods:

"To examine how subjects in a group characterized by low engagement in protective behaviors differed from the average subject, we first calculated z-scores on a number of variables of interest (related to risk perception and affects of the pandemic). We followed this by testing which of these variables predicted membership of the low engagement group versus the rest of the sample using logistic regression."

And the results of the logistic regression are reported in the results section as follows:

"We next used logistic regression to identify variables that uniquely predicted membership of the low engagement group. This revealed three significant predictors, all of which negatively predicted membership of this group: perceived personal severity of potential infection ($\beta = -0.34, p = .04$), reported frequency of searching online for information about the virus ($\beta = -0.31, p = .02$) and feeling personally affected by the pandemic ($\beta = -0.81, p < .001$). Full results of this analysis are shown in Figure 4D and Table S11. Together, the results of this analysis identify a subgroup who reported low engagement in a range of protective behaviors, and who tended to feel they had not been personally affected by the pandemic and were not searching for information about the situation."

b. Did the authors look at the differences between perceived risks for self and for the average person in the state? This would be an index of comparative optimism, which could be more indicative of some self-reported protective behaviors than "pure" risk perceptions.

This is an interesting idea, and we have run additional regressions predicting engagement in these behaviors based on the difference in perceived risk, controlling for age. However, while this showed a significant effect in the primary 75% of our sample, no effect was present in the validation 25%. Therefore, the results are ultimately inconclusive. We have described this in the methods as follows:

"We also performed additional analyses using optimism bias, calculated as the difference between perceived infection probability for the self-versus that for the average person in the state, as a predictor in addition to age as a covariate."

And we have added the results to the results section:

"We also performed separate regressions predicting engagement in these protective behaviors based on levels of optimism bias. This revealed significant effects of optimism bias in the initial 75% of the sample, (handwashing $p = .001$, social distancing $p = .001$), however these were not replicated in the validation sample (handwashing $p = .47$, social distancing $p = .21$)."

c. Btw. do current estimates still suggest that 80% of the population may contract COVID-19 (bottom p. 5)?

Thank you for noticing this. Providing an accurate estimate of the possible number of infections has turned out to be less simple than we may have imagined in March (being a political issue as much as an epidemiological one, particularly in the United States), and estimates provided at the time were likely inaccurate due to the novelty of the virus and limited testing. We have rewritten the relevant sentence to emphasize the uncertainty about this estimate, while still acknowledging that the 80% figure was being reported in the media at the time of data collection.

"Although it is difficult to determine exactly how widespread the pandemic will be in the US, estimates reported in the media when data was collected suggested that up to 80% of the population may contract the disease"

d. It would be nice to read a bit more of a discussion about to what extent the differences between self and average might make sense e.g., the sample was 55% female and largely young, so it might make sense that they don't fear the virus as much, given that they are pretty much the furthest away from being a risk population as possible (this could be briefly done in the discussion)

This is an important point, and we have added the following to the discussion to cover this issue:

"Finally, we also note that the sample were on average at low risk of serious illness due to the pandemic, being majority female and young (18), and therefore the observed optimism bias could be based in being genuinely lower risk than the average person. While we found some evidence that the level of optimism bias was associated with age, this was a weak effect, explaining only 0.5% of the variance in optimism bias."

e. Did the authors look into education level? It seems for example that white-collar workers can more easily reduce physical contact than blue-collar works.

Thank you for this suggestion. While we do not have the necessary data to look at the effects of different types of employment, we are able to split subjects into college educated and non-college educated, which may act as a rough proxy for employment type. This produced some interesting results, showing that individuals without college education generally perceived risk to be lower.

Although this does not directly measure employment, college educated individuals are more likely to work in low-contact, office-based jobs. As a result, if risk perception is a result of genuine risk, college educated individuals should generally perceive risk as lower, which is the opposite to what our results show.

We also saw that non-college educated individuals were more willing to attend events with large numbers of people, which is unlikely to be an employment-based behavior. We have added full details of these analyses to the supplementary material, and have reported them in the results as follows:

"We observed the strongest effects when focusing on education level. Individuals who had completed college-level education or higher reported their personal probability and severity of infection, along with that of the average person in their neighborhood, as higher than those who had not completed college-level education. More striking findings were observed when looking at reported engagement in protective behaviors, where those with college-level education reported engaging in almost all measured protective behaviors to a greater extent than those without a college education."

We have also added the following to the discussion in relation to this issue:

"One aspect of the relationship between perceived risk and protective behavior that we were unable to fully account for is the issue of external factors influencing the ability to engage in protective behavior. The COVID-19 pandemic has affected people differently based on numerous social factors, most notably employment and socioeconomic status, with low-income individuals, often from minority backgrounds, working in environments that necessitate contact with other people (19, 20). Thus, it is possible that variation in perceived risk and reported engagement in protected behavior here may partially depend on these unmeasured factors, as some individuals are at a genuine high risk of infection and are unable to engage in protective behaviors. However, some of our results paint a more complex picture. When examining differences between college educated and non-college educated individuals, we saw that those with a college-level education both perceived their personal risk to be higher and reported engaging in protective behavior to a greater extent. Notably, this included willingness to attend events with large numbers of other people, which is unlikely to be highly dependent on employment in the same way that avoiding social interactions could be. As such, this suggests that individuals with a higher level of education perceive risk as higher and also report engaging in protective behaviors to a greater extent, in a manner that is not purely dependent on employment status."

5) Subgroups/ Gaussian Mixture Model: We have no expertise in this type of modelling so we cannot adequately assess the rigor of this part of the analysis.

First, we wish to inform the reviewers that we noticed a minor error in this analysis during revisions (where data for one of the protective behavior items was coded incorrectly). We have fixed this error in the revised manuscript, and the results remain the same as the prior version, although the exact statistics may be different.

a. Could the authors characterize the clusters a bit more? They provide almost no information (or even discussion about) to what extent a low level of engagement might be justified. Maybe the people who engage little are absolutely justified in doing so and it's just a rational belief.

Thank you for this suggestion, we agree that the characterization of the clusters could have been more detailed. As mentioned previously, we have included an extended analysis predicting group membership from a range of variables. These results do not suggest that lack of engagement is entirely justified. For example, age is not a significant predictor (we might expect younger individuals to feel justified in engaging less), and education level also does not significantly predict group membership (where we may expect that individuals with lower levels of education are less able to avoid social interaction in the course of employment). We have added the following to the discussion to mention this issue:

"Our results suggest that these individuals are not disengaged due to genuine low risk or inability to engage in protective behaviors, being no different in age or education level to the rest of the sample. Instead, membership of this group was predicted by feeling personally unaffected and not seeking information about the pandemic."

b. How many people are in those clusters? Is it a lot of people or just very few? This makes a difference because you'd expect a certain amount of people in each cluster based on random effects, but it would be interesting if e.g., the low-engagement cluster is much larger than would be expected.

Thank you for suggesting this, it is indeed interesting to see how many people fall into each cluster. We have amended Figure 4 to replace the weights of the components (which correspond roughly, but not exactly, to the number of subjects) with the exact number of subjects in each group. There are 122 subjects in the low engagement group, making it the 5th most common. We have also added these numbers to the results sections:

"There were 135 subjects in the low engagement subgroup and 190 in the high engagement subgroup."

c. Are the number of subgroups pre-specified in the analyses?

The Bayesian approach allows the number of subgroups to be determined automatically. Although we place an upper bound on the number (in this case 20), the model-fitting procedure assigns weights to each subgroup, and if the weight is negligible (we set this threshold at a weight of 0.01, where weights range between 0 and 1) we exclude the subgroup. We have added the following to the results to clarify this aspect of the analysis:

"The Bayesian GMM approach allows the number of components to be determined automatically, and this resulted in 16 components being identified."

6) Additional data:

a. Did the authors collect data if restrictions were perceived as forced versus voluntary?

We did not collect this data, unfortunately, although we agree it would be an interesting aspect of risk perception to look at. However, at the time of data collection there had been little in the way of forced engagement in protective behaviors, and the only directions from government agencies were recommendations rather than rules.

b. Did they assess risks for events that are unrelated to COVID-19 to get some baseline for risk perception?

We did not measure any other form of risk perception, although this is an interesting suggestion. We have added the following to the discussion to cover the issue of baseline risk perception on our measures:

"We also note that we did not assess general risk perception levels, and so we cannot determine the extent to which this bias is specific to the pandemic."

c. Did the authors assess any links between optimism bias and increased coping mechanism/better mental health?

We did not collect detailed assessments of coping mechanisms or mental health, and so we are not able to answer these questions fully. We did collect data on state and trait anxiety levels, however this will be analyzed thoroughly in forthcoming studies.

d. It seems quite likely that the authors continue to collect data. If so, it would be nice if they could hint at this in the discussion. Readers could thus look out for new publications.

We are indeed still collecting data, and we have now mentioned this in the discussion as follows:

"These results provide insights into the development of psychological and behavioral responses to the early stages of a pandemic, and we continue to collect additional data to determine how these responses evolves as the pandemic progresses."

7) Some further points

a. Probability & likelihood are not the same statistically. Please try to be as precise as possible.

Thank you for noting this – we have replaced references to likelihood with probability where appropriate. Our items used the word "likelihood" as it is generally interpreted as having the same meaning as probability outside statistical contexts however, and we have added the following to clarify this:

"We used the term "likelihood" in the items as it is colloquially interpreted as representing probability, however we assume that responses refer to probability of the event occurring in the formal statistical sense"

b. The authors can only analyze self-reported behavior. This is mentioned and it' also obvious from the study design but it could be highlighted a bit more, for example in the title: "self-reported protective behavior"

Thank you for this suggestion. We have changed the title as suggested and have added further mentions of the fact that behavior is self-reported throughout the manuscript, which are highlighted in the revised version.

c. Please add a reference for claim about when which restrictions were made in US relative to data collection (p. 4/14 bottom)

Thank you for noting this. We have included a reference for these dates. We have also clarified the nature of these announcements, being guidance rather than enforced restrictions.

Reviewer: 2

Comments to the Author(s)

1) It would be better to place the figure 1 in the section of results so that the descriptions in the text and figure would align.

Thank you for noting this, we have moved the figure to the results as suggested so that it lines up with the relevant text.

2) In the study design, can you justify the combined cross sectional and longitudinal study along with its uses in the study?

Thank you for this suggestion, we agree that the use of both approaches needs to be motivated further. While the cross-sectional analysis was sufficient to test many of our hypotheses, we performed the longitudinal study to confirm that the differences we saw over time in separate samples were genuine within-subject changes, as opposed to differences between samples collected on different dates. We have added the following sections to the method and results to clarify this:

"Second, we used repeated ANOVAs in our longitudinal dataset to test for within-subject changes over time. Here, the factors were the same as above, however date of testing was a within-subject factor in this case. This allowed us to confirm whether differences in samples tested across different days did indeed represent a true change over time."

"In between subjects analyses, perceived probability of infection differed across samples tested on different days, demonstrating a higher rate in subjects tested on later dates ($F(6, 1579) = 6.48, p < .001, \eta_p^2 = 0.024$, Figure 2A). To confirm that represented a change over time, we examined increases in perceived probability within-subjects in a subsample followed up after 5 days, finding that this did indeed increase over time"

"Importantly, we verified this pattern of change over time using repeated testing of a subset of our sample."

3) Please review the results with more descriptions and explanations.

We have revised the results section to include more description and explanation of the relevant results by including summaries of the main findings throughout:

"Together, these results indicate a clear pattern of optimistic risk perception, with subjects rating themselves as being at lower risk of infection than the average person, which changed rapidly over the period of the study."

"Together, these results indicate that subjects generally reported engaging in protective behaviors, and that their level of engagement increased dramatically over in the five days during which the study was conducted."

"These results show that individuals who perceived their probability of becoming infected and believed they would be strongly affected by the broader consequences of the pandemic tended to engage in protective behaviors to a greater extent."

"Together, the results of this analysis identify a subgroup who reported low engagement in a range of protective behaviors, and who tended to feel they had not been personally affected by the pandemic and were not searching for information about the situation."

4) The discussion should focus on the results with evidences.

We have revised the discussion as suggested to include a clear summary of the results, with the relevant evidence referenced, as follows:

"Our primary analyses focused on levels of risk perception and self-reported engagement in protective behavior over time. We found that subjects tended to perceive their personal risk of infection as being higher than the average person in their neighborhood, state, and country, and that reported risk increased between the dates of the study (3/11/20 to 3/16/20). We also found that, on average, subjects reported engaging in many forms of protective behavior, such as hand-washing and social distancing, and reported engagement increased dramatically over the course of the study. Importantly, we verified this pattern of change over time seen in between-subject analyses using repeated testing of a subset of our sample. We additionally investigated predictors of reported engagement in protective behaviors, finding that the clearest predictors were perceived personal risk of infection and being affected by the broader societal consequences of the pandemic. Finally, we sought to identify subgroups of individuals based on their reported engagement in protective behaviors. This analysis identified a subgroup who reported low engagement in all the behaviors studied, and who tended to feel personally unaffected by the pandemic and who were not searching for information about the virus."

Reviewer: 3

Comments to the Author(s)

This manuscript is concerned with perceptions of risk to the self and to others relating to the current COVID-19 global pandemic and associated protective actions.

Overall I think this is an exemplary, thoughtful and timely piece of applied research. A couple of the figures are a little bit busy but they are also good at clearly presenting a lot of information so I think it's a worthwhile tradeoff to make. The one place I think could use a little more attention would be the discussion section, there are a couple of places where some additional theory could help to bolster or provide potential explanations for some of the results as well as add nuance to some of the recommendations. With minor revisions of this kind I would recommend publication of this manuscript.

Thank you for these positive comments, in particular for pointing us towards highly relevant literature that we had not referenced in the prior version of the manuscript.

Methods:

One thing that came to mind was whether there were any problematic intercorrelations between the risk perception items, was there any evidence of high intercorrelations that might have masked some of the results in the linear regressions?

Thank you for this suggestion. We have checked the variance inflation factors for each variable included in the regression models to determine whether any showed problematic levels of multicollinearity. All values were below 5, which is suggested as an indication of high multicollinearity. We have described this in the methods as follows:

"We checked for multicollinearity between predictor variables using the variance inflation factor. All values were below 5, indicating no problematic multicollinearity between predictors (15)."

Results and Discussion:

Another thing I think is very interesting is looking at the disengaged group. While this group doesn't appear to differ by gender (which might be expected if it's the result of something like the "white male effect" as in Finucane et al, 2000), I think there might be something to the idea that this group is driven by other cultural or socio-political motivations. While obviously you do not necessarily have to data to conclusively rule out or support something like a cultural cognition argument, it might be worth mentioning some of that literature (for instance the Kahan et al, 2011 below) in the discussion. It seems like there are some similar points to be made about scientific consensus with regards to the pandemic that are routinely made with respect to scientific consensus on climate change. I'm having a little trouble following the Bayesian analysis myself but I would be interested to see if elevated concerns with respect to economic impacts from the global response to the virus might characterize one group over another and whether a cultural argument could be made for such differences. To be clear, I don't necessarily think

that this requires a change to the manuscript, but I think some of these issues might help to add nuance to the discussion, particularly with reference to theory beyond the optimism bias argument.

Thank you for this suggestion, we agree that the characteristics of this low engagement group are particularly interesting.

In terms of analysis, we have performed an additional logistic regression analysis predicting membership of the low engagement group from a number of factors, including perceived risk from the global economic consequences of the pandemic. Interestingly, the only significant (negative) predictors were perceived severity of illness if infected, having been personally affected by the pandemic (in terms of work, school etc.) and searching for information about the pandemic online. Thus, it seems that concerns about the broader economic effects of the pandemic (or lack thereof) were not a differentiating feature of this group. These results are described as follows in the manuscript:

"We next used logistic regression to identify variables that uniquely predicted membership of the low engagement group. This revealed two significant predictors, both of which negatively predicted membership of this group: reported frequency of searching online for information about the virus ($\beta = -0.31, p = .02$) and feeling personally affected by the pandemic ($\beta = -0.29, p < .001$). Full results of this analysis are shown in Figure 4D and Table S11."

In terms of discussion, the point about cultural cognition and general influences of culture is an interesting one, although sadly one that we do not have the data to investigate properly. As the united states is such a diverse country, there are obviously many cultural factors that may influence differences in risk perception. We have added the following to the discussion to address this issue:

"We also did not address cultural factors that may play a key role in perceptions of risk and engagement in protective behaviors. In other areas where accurate public risk perception and behavior is required to enable broad beneficial societal change, such as climate change, it has been established that responses can differ dramatically depending on the individual's cultural environment, a phenomenon known as cultural cognition (24). This prior work has demonstrated that individuals in certain cultures can perceive risk in a way that differs substantially from expert consensus, and this phenomenon may explain the low levels of perceived risk seen here in a subgroup of subjects. Indeed, throughout the pandemic, adherence to public health measures has been found to depend on political affiliation (25), providing some evidence for this hypothesis. Additionally, this study was conducted in the very early stages of the pandemic in the United States, at a time where scientific knowledge about the virus was limited. In the absence of a clear scientific consensus, cultural influences may be exaggerated."

Another potential source of nuance in the discussion might come from fear appeal literature such as that from Protection Motivation Theory (Rogers, 1975; Maddux & Rogers, 1983) and the Extended Parallel Process model (Witte, 1992), both of which suggest that substantive change in cognitions related to protective action in response to fear appeals (in this case information from government and media sources concerning covid and the appropriate responses) will be the result of both high risk perception and high self- and response efficacy (i.e. "There's something I can do" and "If I do it, it will work"). In particular, the extended parallel process model would suggest that refraining from protective action

might be the result of one of two potential processes, either because risk perception is initially low (resulting in no action), or risk perception is high but one or more efficacy variable is low leading to a motivation to control fear as opposed to controlling danger which can lead to a reduction in risk perception (that is, if I am scared but there's nothing I think I can do that will help, I downplay the risk to make myself more comfortable). These may also be worth considering when thinking about how to engage the disengaged group beyond additional education efforts in the discussion.

Thank you for these insightful suggestions. We agree that these theoretical perspectives can help contextualize our results, and they can inform strategies to engage individuals in protective behavior. The role of perceived efficacy is particularly interesting given widespread beliefs about the lack of efficacy of mask wearing and social distancing in the United States (related to the point above). We have added the following paragraph to the discussion to cover these issues:

"These implications should be considered in the context of prior work on the role of fear in motivating beneficial behaviors, however. In particular, one factor that has been frequently highlighted in the literature is self-efficacy. In the presence of a known threat, protective behaviors are most likely to be performed if the individual feels these behaviors will be effective in reducing risk (20) and they will be able to successfully perform the actions necessary to reduce the risk (21). Similarly, other work emphasizes the importance of engaging in protective behaviors as a response to perceived risk as opposed to fear, and this depends on perceived efficacy (22). That is, responses to a threat in the context of low perceived efficacy will lead to efforts to reduce fear, such as denial and other coping strategies, while high perceived efficacy will lead to the adoption of behaviors to reduce the risk from the threat itself. One recent study has demonstrated that perceived efficacy predicts self-reported engagement in protective behaviors during the COVID-19 pandemic (23), indicating that this is an area worthy of consideration in efforts to motivate engagement in protective behavior."

Finucane, M. L., Slovic, P., Mertz, C. K., Flynn, J., & Satterfield, T. A. (2000). Gender, race, and perceived risk: The 'white male' effect. *Health, risk & society*, 2(2), 159-172.

Kahan, D. M., Jenkins-Smith, H., & Braman, D. (2011). Cultural cognition of scientific consensus. *Journal of risk research*, 14(2), 147-174.

Maddux, J. E., & Rogers, R. W. (1983). Protection motivation and self-efficacy: A revised theory of fear appeals and attitude change. *Journal of experimental social psychology*, 19(5), 469-479.

Rogers, R. W. (1975). A protection motivation theory of fear appeals and attitude change. *The journal of psychology*, 91(1), 93-114.

Witte, K. (1992). Putting the fear back into fear appeals: The extended parallel process model. *Communications Monographs*, 59(4), 329-349.

Minor General Notes

Page 2 Line 51: I believe this sentence should read "we are not aware of any data..." or "we are aware of no data..."

Page 3 Line 50: "survey" should probably be pluralized

Page 5 Line 31: "the a multivariate distribution" should use either "the" or "a" but not both

Page 8 Line 58: I believe this should be "Bayesian Gaussian mixture model"

Thank you for noting these issues, we have corrected them in the revised version of the manuscript.